# MODEL METAMERS REVEAL INVARIANCES IN GRAPH NEURAL NETWORKS

## ABSTRACT

In recent years, deep neural networks have been extensively employed in perceptual systems to learn representations endowed with invariances, aiming to emulate the invariance mechanisms observed in the human brain. However, studies in the visual and auditory domains have confirmed that significant gaps remain between the invariance properties of artificial neural networks and those of humans. To investigate the invariance behavior within graph neural networks (GNNs), we introduce a model "metamers" generation technique. By optimizing input graphs such that their internal node activations match those of a reference graph, we obtain graphs that are equivalent in the model's representation space, yet differ significantly in both structure and node features. Our theoretical analysis focuses on two aspects: the local metamer dimension for a single node and the activation-induced volume change of the metamer manifold. Utilizing this approach, we uncover extreme levels of representational invariance across several classic GNN architectures. Although targeted architectural and training adjustments can partially reduce this excessive invariance, they do not fundamentally resolve it. Finally, we quantify the deviation between metamer graphs and their original counterparts, revealing unique failure modes of current GNNs and providing a complementary benchmark for model evaluation.

## 1 INTRODUCTION

In neuroscience, a core objective is to build perceptual models that replicate both the responses and behaviors of the brain Kell et al. (2018); Schrimpf et al. (2020). Inspired by the hierarchical structure of biological sensory systems, modern neural networks transform raw inputs into task-relevant representations and have become the dominant framework for modeling perception Richards et al. (2019). A prevailing hypothesis suggests that optimizing these networks for recognition tasks will naturally induce human-like invariances—robustness to irrelevant variations such as pose, lighting, or speaker identity. While much attention has been paid to models failing under minor perturbations that humans easily tolerate, less discussed are cases where models remain stable under distortions that render inputs unrecognizable to humans. These instances highlight a different issue: the invariances learned by neural networks can diverge sharply from those of human perception, a point discussed by Feather et al. (2023) in their analysis of model–human mismatch.

Figure 1 conceptualizes the universe of all possible inputs as a single "stimulus space", in which the green-shaded region marks every input that humans would subjectively judge to belong to the same category as a given reference sample, and the yellow-shaded region marks every input that the model's output assigns to that same category. When either of these regions contains inputs that look obviously different from the reference on the surface yet evoke identical internal representations—whether in human neural activity or the model's intermediate activations—and are still classified as the same category, we call those inputs "metamers" (illustrated by the shaded circles). By comparing the size and overlap of the human and model metamer regions, we obtain an intuitive measure of how closely the model's learned invariances match those of human perception.

Building on this perspective, we propose graph model metamers as a tool to investigate invariance in graph neural networks (GNNs) Yi et al. (2025); Xu et al. (2025). As shown in Figure 2, given a reference graph $G$, we synthesize a graph $G'$ that matches its internal representation at a chosen GNN layer, while allowing node features and/or structure to vary.

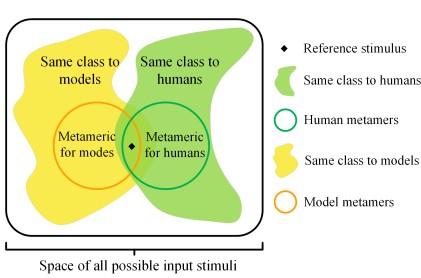

Figure 1: Overlap between human and model metamers in input space.

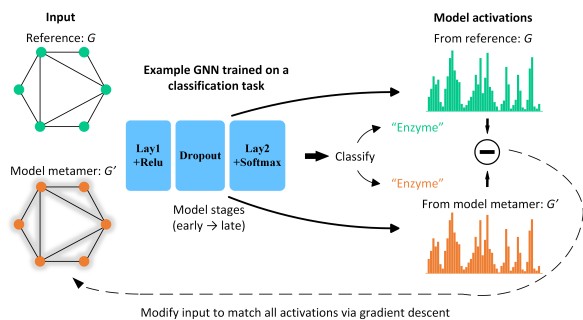

Figure 2: Metamer generation for GNNs.

GNNs achieve strong results across domains but face four core limitations. Their expressiveness matches the 1 WL test and fail to distinguish certain graph patterns Wang & Zhang (2022); K hop propagation Feng et al. (2022) and structure aware designs Wijesinghe & Wang (2022) effectively alleviate this limitation. Depth induces oversmoothing, eased by residual connections Scholkemper et al. (2025) and reverse message passing Park et al. (2024). Sparse connectivity causes oversquashing; multi track routing Pei et al. (2024) and non dissipative updates Gravina et al. (2025) preserve long range signals. Under heterophily, standard GNNs underperform; specialized architectures and graph rewiring help Chen et al. (2024); Bi et al. (2024); Yang et al. (2025).

Despite progress, it remains unclear whether GNN learned invariances are appropriate for reliable predictions. Maron et al. (2019) characterize the full family of permutation invariant and equivariant linear layers, give explicit bases whose sizes follow Bell numbers, and use these results to design symmetry respecting architectures. Our objective is different: rather than designing layers, we audit trained models a posteriori by searching for inputs with distinct structures and features that nonetheless yield identical internal activations, thereby revealing over invariance that emerges during learning. Prior expressiveness studies ask whether models can in theory separate all non-isomorphic graphs Joshi et al. (2023); Bouritsas et al. (2023), whereas we examine what current networks actually do; explainability work emphasizes local feature attribution Azzolin et al. (2025); Gui et al. (2024), while we analyze invariances of internal representations under controlled input variation.

To investigate invariance in GNNs, we propose a metamer-based framework. This approach exposes a high degree of representational invariance in standard GNNs. To address this, we introduce architectural and training modifications that mitigate the effect. We uncover a characteristic failure mode, and provide a new benchmark for evaluation. The main contributions of this work are:

- We theoretically characterize metamers via the nodewise local metamer dimension and the activation induced volume change of the metamer manifold (Appendix).

- We are the first to introduce metamer generation for GNNs, revealing that standard architectures exhibit pronounced over invariance in their internal representations (Section 3).

- By quantifying each metamer's deviation from its source graph, we uncover a distinctive failure mode of contemporary GNNs and establish a complementary benchmark for model evaluation (Section 4).

- We propose targeted architectural and training modifications across five canonical GNN variants to effectively mitigate this excessive invariance (Section 4).

## 2 INVARIANCES IN SENSORY MODELS AND THE HUMAN PERCEPTUAL SYSTEM

Feather et al. (2023) proposed and validated the model metamer approach to compare invariance in artificial models and human perception. A model metamer is a synthetic input optimized to match the internal activations of a reference sample at a specific network layer. The method was applied to visual and auditory models to assess the emergence of human-like invariances across layers.

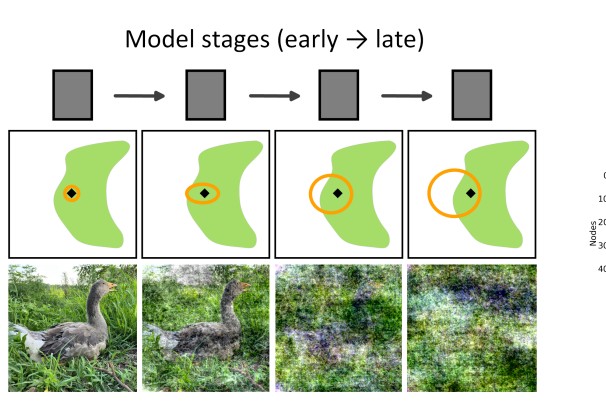

Figure 3: Metamers generated from deeper layers of the model become increasingly unrecognizable to humans.

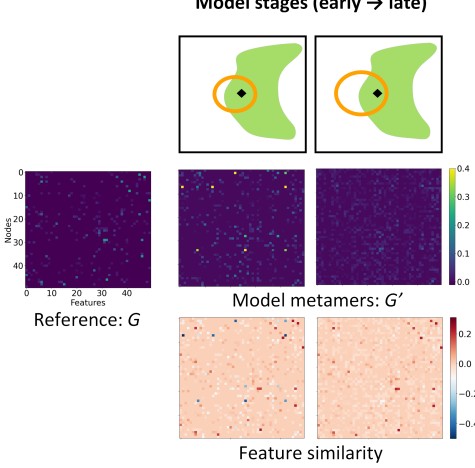

Figure 4: Metamer generation for GNNs.

As shown in Figure 3, in the visual domain, Feather et al. (2023) evaluated over a dozen architectures (e.g., AlexNet, VGG-19, ResNet-50) on a 16-class object classification task. Metamers were generated from successive layers, and human participants attempted to classify them. While early-layer metamers remained somewhat interpretable, those from deeper layers resembled noise, yielding near-chance accuracy—suggesting a divergence from human visual invariance. In the auditory domain, two cochleagram-based models were tested on a 793-class word task, with human accuracy similarly collapsing on deep-layer metamers, again revealing a mismatch with human perceptual invariance.

The authors evaluated several strategies—such as self-supervised learning, stylized ImageNet, low-pass filtering, and adversarial training—for their effect on metamer recognizability. Adversarial training yielded the most improvement, though none fully bridged the gap between model and human invariance. These findings highlight a systematic mismatch in current models and provide a general benchmark for aligning them with human perception.

## 3 GRAPH MODELS METAMERS GENERATION

In this section, we focus on the generation of metamers for GNNs, including both feature-based and structure-based metamers. We detail the generation process and introduce several methods aimed at reducing the extent of the metamer manifold. In addition, we design evaluation metrics to quantify the invariance exhibited by GNNs. We begin with necessary preliminaries.

### 3.1 PRELIMINARIES

**Notations.** Given a graph $G = (V, E, X) \in \mathcal{G}$ with $n = |V|$ nodes, edge set $E$, and $\mathcal{G}$ are graph sets, node features $X \in \mathbb{R}^{n \times d}$ where $x_v \in \mathbb{R}^d$ denotes the feature of node $v$, the adjacency matrix $A \in \mathbb{R}^{n \times n}$ encodes edge connectivity. In a neighbor-aggregation GNN, the $k$-th layer updates each node's representation by aggregating features from its neighbors:

$$\tilde{h}_v^{(k)} = \sum_{u \in \mathcal{N}(v)} \alpha_{uv} \, \psi^{(k)}\big(h_v^{(k-1)}, h_u^{(k-1)}\big) \tag{1}$$

$$h_v^{(k)} = \sigma\Big(\mathbf{W}^{(k)} \, [\, h_v^{(k-1)} \, \| \, \tilde{h}_v^{(k)} \,]\Big) \tag{2}$$

where $\tilde{h}_v^{(k)}$ is the aggregated message at layer $k$, $h_v^{(k-1)}$ the previous-layer embedding, $h_v^{(k)}$ the updated embedding, and $h_v^{(0)} = x_v$; $\mathcal{N}(v)$ the neighbors of $v$, $\alpha_{uv}$ normalized edge weights, $\psi^{(k)}$ the message function, $\mathbf{W}^{(k)}$ the weight matrix, and $[\cdot \| \cdot]$ concatenation operator.

**Graph model metamer.** Let $f : \{\mathcal{G} \to \mathbb{R}^m\}$ denote our GNN-based graph embedding function, which may represent one or multiple layers of the GNN model. In this framework, the input graph $G$ is regarded as a stimulus, and the corresponding output $f(G)$ produced by the embedding function is referred to as the activation. For a given reference graph $G$, we define its graph metamer set as follows:

$$\mathcal{M}_f(G) \;=\; \big\{\, G' \in \mathcal{G} \mid f(G') \approx f(G) \big\} \tag{3}$$

In practice, $G'$ is a graph model metamer of $G$, $\mathcal{M}_f(G)$ reflects the model's invariance: its volume indicates the extent of perturbations that preserve the embedding. A larger $\mathcal{M}_f(G)$ suggests stronger invariance but reduced discriminability, while a smaller one implies greater sensitivity.

### 3.2 METAMER GENERATION OBJECTIVE AND OPTIMIZATION

Figure 2 provides an illustration of the objective and optimization process used for metamer generation. Given a reference graph $G$, our goal is to synthesize a metamer $G'$ that produces an identical internal representation at a selected layer of a pretrained GNN, while allowing free variation in the graph structure and/or node features. Let the GNN define the following mapping:

$$f : G \longmapsto \big\{\, h^{(1)}, h^{(2)}, \ldots, h^{(K)}, \, y \big\} \tag{4}$$

where $h^{(k)} \in \mathbb{R}^{d_k}$ denotes the activation vector at layer $k$, and $y$ represents the final output of the model (e.g., a class label).

We define the activation matching loss:

$$\mathcal{L}_{\text{act}} = \frac{\big\| h'^{(k)} - h^{(k)} \big\|_2^2}{\big\| h^{(k)} \big\|_2^2} \tag{5}$$

which penalizes any deviation of the synthesized graph's $k$-th layer activation from that of the reference graph. Here, $k$ is manually selected to target a specific layer of the GNN, allowing us to investigate the model's invariance properties at different levels of representation.

The synthesis of a metamer graph $G'$ begins by initializing $G'_0$ with either random node features (Section 3.2 for details) or random edge connections (Section 3.3 for details). Since most GNNs do not use edge features, we do not consider metamer generation in the edge-feature space. Gradient-based updates are applied until convergence, with the process terminating after $T$ steps. At each iteration $t$, we compute the gradient of the loss $\mathcal{L}_{\text{act}}$ with respect to the current graph input $G'_t$, and perform the following update:

$$G'_{t+1} = \text{Proj}(G'_t - \eta \, \nabla_{G'} \, \mathcal{L}(G'_t)) \tag{6}$$

where $\eta$ denotes the learning rate, and $\text{Proj}$ represents a projection operator that enforces validity constraints on the graph (e.g., clipping node features to the range $[0, 1]$, or thresholding continuous edge weights to obtain a valid adjacency matrix). After $T$ iterations, the resulting graph $G'_T$ is considered the synthesized metamer for layer $k$. This procedure ensures that $G'_T$ produces a similar internal activation at layer $k$ to that of the reference graph $G$, while allowing for maximal variation in other aspects of the input, subject to the imposed constraints.

### 3.3 METAMER CONSTRUCTION: NODE FEATURE GENERATION

The construction of a metamer graph $G'$ consists of generating the node feature matrix $X'$ and the adjacency matrix $A'$. We begin by introducing the process for generating $X'$. The initialization of $X'$ is based on the mean $\mu \in \mathbb{R}^d$ and standard deviation $\tau \in \mathbb{R}^d$ of the reference graph $G$'s feature matrix, ensuring that the synthesized features remain within a comparable distribution.

$$X'_{\text{soft}} = \mu \, \mathbf{1}_{|V|}^{\top} \; + \; \tau \; \odot \; \epsilon, \quad \epsilon \sim \mathcal{N}(0, 1)^{|V| \times d} \tag{7}$$

where $\mathbf{1}_N \in \mathbb{R}^N$ is the all-ones vector, "$\odot$" denotes elementwise multiplication with broadcasting, and $\epsilon \in \mathbb{R}^{|V| \times d}$ are independent and identically distributed according to the standard normal distribution $\mathcal{N}(0, 1)$. Since our experimental datasets include binarized features, we next apply discretization to the subset of features that require it, using binary-valued features as a representative example.

In the forward pass, we first apply an elementwise sigmoid function with slope parameter $s > 0$ to obtain a soft probability matrix:

$$P = \sigma\big(s\, X'_{\text{soft}}\big) \in (0,1)^{|V| \times d} \tag{8}$$

The matrix $P$ represents elementwise "soft" probabilities of activation for binary features. To derive a discrete mask, we enforce a target sparsity $\rho \in (0,1)$—a learnable scalar initialized to the empirical density of the reference feature matrix $X$—which specifies the desired fraction of active entries. We then identify the top $\rho$-fraction of entries in $P$ to form a hard binary mask $X'_{\text{hard}}$. To reconcile discreteness in the forward pass with differentiability in the backward pass, we employ the straight-through estimator (STE) Bengio et al. (2013):

$$X' = P + \big(X'_{\text{hard}} - P\big)\big|_{\text{stopgrad}} \tag{9}$$

where $\big(X'_{\text{hard}} - P\big)\big|_{\text{stopgrad}}$ contributes zero gradient, so that during backpropagation gradients flow purely through $P$. As a result, $X'$ is exactly binary (equal to $X'_{\text{hard}}$) at inference time, while remaining end-to-end trainable via the continuous surrogate $P$. Finally, we include a margin regularization term:

$$\mathcal{L}_{\text{margin}} = \lambda_{\text{reg}}\, \frac{1}{|V|\, d} \sum_{v=1}^{|V|} \sum_{u=1}^{d} P_{v,u}\big(1 - P_{v,u}\big) \tag{10}$$

to prevent $P$ from collapsing to the extremes 0 or 1 prematurely. Moreover, continuous feature generation is straightforward: we apply ReLU to the soft features $X'_{\text{soft}}$ and iteratively optimize it to ensure all entries remain strictly positive. As shown in Figure 4, the second row presents metamer features generated from different layers of the model, while the third row displays the similarity between these metamer features and the original features $X$ of the reference graph $G$. As the layer depth increases, the difference between $X'$ and $X$ becomes increasingly pronounced. The first row further illustrates that the metamers progressively deviate from human-recognizable patterns as they are derived from deeper layers.

## 3.4 METAMER CONSTRUCTION: STRUCTURE GENERATION

For adjacency mask generation, we initialize a continuous adjacency parameter $A'_{\text{soft}} \in \mathbb{R}^{|V| \times |V|}$ by sampling a random upper-triangular matrix and symmetrizing it, ensuring that no self-loops are present on the diagonal. In the forward pass, we compute a soft adjacency probability matrix as follows:

$$P = \sigma\big(s\, A'_{\text{soft}}\big) \in (0,1)^{|V| \times |V|} \tag{11}$$

using the same sigmoid slope $s > 0$. A learnable scalar $\rho \in (0,1)$—initialized to the empirical edge density of the reference adjacency matrix $A$—specifies the target fraction of edges. We then select the top $\rho$-fraction of entries in $P$ to construct a hard adjacency mask $A'_{\text{hard}}$, and use the STE during backpropagation.

$$A = P + \big(A'_{\text{hard}} - P\big)\big|_{\text{stopgrad}} \tag{12}$$

This ensures that $A$ is strictly binary during the forward pass, while gradients are allowed to flow through the soft matrix $P$ during backpropagation. The resulting symmetric binary matrix $A \in {0,1}^{|V| \times |V|}$ serves as the adjacency mask for subsequent graph convolution or message-passing layers.

## 3.5 QUANTITATIVE METRICS FOR GNN INVARIANCE

To evaluate the invariance of a GNN in node classification tasks, we introduce a consistency objective that jointly accounts for feature-level similarity, structural similarity, and classification agreement.

For feature-based metamers—where the graph structure is fixed—we measure the similarity between the generated features $X'$ and the reference features $X$ using cosine similarity. Since graph features are high-dimensional and not directly interpretable, cosine similarity serves as a distribution-aware proxy. If the GNN yields identical outputs for inputs with substantially different feature distributions, this indicates an overly permissive invariance that may not align with human intuition.

$$S_{\text{feat}} = \frac{\langle X,\, X' \rangle}{\|X\|\, \|X'\|} \in [-1, 1] \tag{13}$$

We also compute the classification match ratio:

$$S_{\text{match}} = \frac{1}{|V|} \sum_{v \in V} \mathbf{1}\big(y_v = y'_v\big) \tag{14}$$

where $y_v$ and $y'_v$ are the predicted labels on the original and feature-metamer graphs, respectively. The feature consistency score ($CS$) is then defined as:

$$CS_{\text{feat}} = S_{\text{feat}}\, S_{\text{match}} + (1 - S_{\text{feat}})(1 - S_{\text{match}}) \tag{15}$$

This score is high when cosine similarity and classification agreement are aligned—either both high (indicating invariance to similar inputs) or both low (indicating sensitivity to dissimilar inputs)—both of which reflect appropriate invariance behavior.

For structure-based metamers, we employ the Weisfeiler–Lehman (WL) graph kernel with degree-based initialization. By iteratively aggregating neighborhood labels, WL captures higher-order sub-tree patterns and yields a similarity score:

$$S_{\text{struct}} = \kappa_{\text{WL}}\big(A,\, A'\big) \in [0, 1] \tag{16}$$

The structure $CS$ is then:

$$CS_{\text{struct}} = S_{\text{struct}}\, S_{\text{match}} + (1 - S_{\text{struct}})(1 - S_{\text{match}}) \tag{17}$$

Similar to the feature-based case, a high structural consistency score indicates that structural similarity aligns with classification agreement.

## 4 EXPERIMENTS

This section evaluates model invariance across diverse graph datasets, outlines the experimental settings and baselines, and examines layer-wise invariance trends along with mitigation strategies.

### 4.1 EXPERIMENTAL SETUP

**Datasets.** We evaluate five node classification datasets: Cora, CiteSeer, and PubMed are homophilic graphs Fowler (2006), while Squirrel and Chameleon are heterophilic Rozemberczki et al. (2021). PubMed is the only dataset with continuous node features; the rest use discrete inputs. This diversity in structure and feature type supports a comprehensive analysis of GNN invariance.

**Setting-up.** All experiments are conducted on a machine equipped with an NVIDIA RTX H200 GPU. We use the Adam optimizer, with a learning rate of 0.001 for GNN training and 0.0005 for metamer generation. All datasets and baseline models are implemented using the PyTorch Geometric library.

**Baselines.** We evaluate invariance across six representative GNN architectures: GCN (spatial convolution) Kipf & Welling (2017), ChebNet (spectral filtering) Defferrard et al. (2016), GraphSAGE (inductive aggregation) Hamilton et al. (2017), GAT (attention-based aggregation) Velickovic et al. (2018), GIN (injective neighborhood functions) Xu et al. (2019), and Graphormer (transformer-based graph modeling) Ying et al. (2021). These models span the core design paradigms of GNNs and serve as the foundation for many contemporary variants.

### 4.2 ANALYZING INVARIANCE VIA FEATURE METAMERS

We evaluated six GNNs on five node-classification datasets via feature-metamer generation, fixing each graph's adjacency and targeting first-layer activations. For each metamer, we measured classification match rate $S_{\text{match}}$ and cosine similarity $S_{\text{feat}}$, combining them into a consistency score $CS_{\text{feat}}$. Table 1 reports the mean and standard deviation over five runs, with $(S_{\text{feat}}, S_{\text{match}})$ shown on the first line of each cell and the resulting consistency score $CS_{\text{feat}}$ on the second.

As discussed in Appendix A.1.1, the dimensionality of the metamer manifold is given by $d-r$, where $d$ is the input feature dimension and $r$ is the rank of the model's Jacobian. Thus, PubMed—having

Table 1: Invariances of GNNs on feature-metamers across datasets.

| Method | Cora | CiteSeer | PubMed | Squirrel | Chameleon |
|---|---|---|---|---|---|
| GCN | 11.4±0.05  96.4±0.32
14.12±0.29 | 8.0±0.08  96.1±0.21
11.25±0.24 | 22.2±0.34  99.9±0.01
22.31±0.34 | 5.0±0.19  84.3±0.93
19.16±1.10 | 7.5±0.42  88.8±0.95
17.00±0.98 |
| ChebNet | 14.8±0.11  96.0±0.32
17.60±0.21 | 10.6±0.15  95.0±0.21
14.48±0.25 | 24.3±0.19  99.9±0.02
24.37±0.20 | 5.0±0.10  86.3±0.24
17.36±0.19 | 10.1±0.67  91.8±0.73
16.69±0.60 |
| GraphSAGE | 12.6±0.06  95.9±0.51
15.61±0.39 | 10.0±0.17  94.7±0.50
14.22±0.49 | 21.2±0.47  99.9±0.02
21.31±0.47 | 5.8±0.09  89.4±0.45
15.13±0.38 | 5.4±0.09  91.4±0.52
13.09±0.50 |
| GIN | 10.4±0.14  97.1±0.23
12.65±0.19 | 9.1±0.16  96.3±0.46
12.16±0.34 | 19.3±0.81  99.8±0.02
19.38±0.80 | 2.3±0.08  90.9±0.79
10.93±0.76 | 2.3±0.21  87.5±1.55
14.19±1.36 |
| GAT | 15.0±0.37  95.0±0.15
18.47±0.26 | 12.8±0.66  94.0±1.08
17.21±1.22 | 41.9±0.64  99.9±0.02
41.94±0.64 | 32.4±1.11  81.7±0.29
38.83±0.76 | 18.8±0.97  85.6±1.09
27.81±0.99 |
| Graphormer | 14.8±0.29  94.6±0.49
**18.61±0.56** | 14.0±0.52  93.2±0.42
**18.88±0.71** | 42.7±0.41  99.8±0.01
**42.80±0.41** | 46.3±1.22  82.8±0.30
**47.63±0.78** | 19.7±0.81  86.1±0.52
**28.14±0.80** |

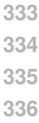

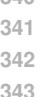

Figure 5: Structure-metamer evaluation on GCN and ChebNet.

the smallest $d$—yields a smaller manifold and lower observed invariance, although its continuous features facilitate optimization and drive $S_{\text{match}}$ toward 100%. In contrast, heterophilic datasets—where baseline accuracy is already low—produce metamers with reduced $S_{\text{match}}$. Notably, GAT and Graphormer, which employ learnable adjacency weights, increase $r$, shrink the local manifold dimension $(d - r)$, and achieve higher $CS_{\text{feat}}$, explaining their superior performance. Although GIN is an injective neural network, Davidson & Dym (2025) have shown that it cannot effectively separate two distinct representations.

## 4.3 Analyzing Invariance via Structural Metamers

We conducted structure-metamer generation experiments on GCN and ChebNet, with results shown in Figure 5. Although the structural similarity scores $S_{\text{struct}}$, computed using the WL kernel, are close to 100%, the classification match scores $S_{\text{match}}$ are only around 80%. This indicates that even small changes in graph structure can significantly alter the GNN's activations, suggesting that structure metamers do not exist for these models.

## 4.4 Mitigating Model Invariance

**Changing the model structure or training method.** We experimented with three strategies to mitigate model over-invariance: replacing the ReLU activation with ELU, applying adversarial training, and adding residual connections. Replacing ReLU with ELU is motivated by the analysis in Appendix A.1.2, where we show that ReLU collapses at zero and introduces additional metamer directions. In contrast, ELU maintains similar expressiveness while avoiding such collapse. Both adversarial training and residual connections aim to increase the rank of the model's Jacobian, thereby reducing the dimensionality of the metamer space and improving sensitivity to input variations. As shown in Figure 6, we evaluated all three strategies across six models and five datasets and recorded the feature-level consistency score $CS_{\text{feat}}$. All methods consistently reduced over-invariance, with adversarial training showing the most stable improvements.

**Increase hidden layer dimension.** Additionally, increasing the dimensionality of the model's hidden (activation) layer can expand the Jacobian matrix, thereby increasing its rank and mitigating over-invariance. As shown in Figure 7, we conducted experiments on four models across five datasets, testing hidden dimensions of 16, 32, and 64. The results show that the $CS_{\text{feat}}$ increases consistently with larger hidden dimensions.

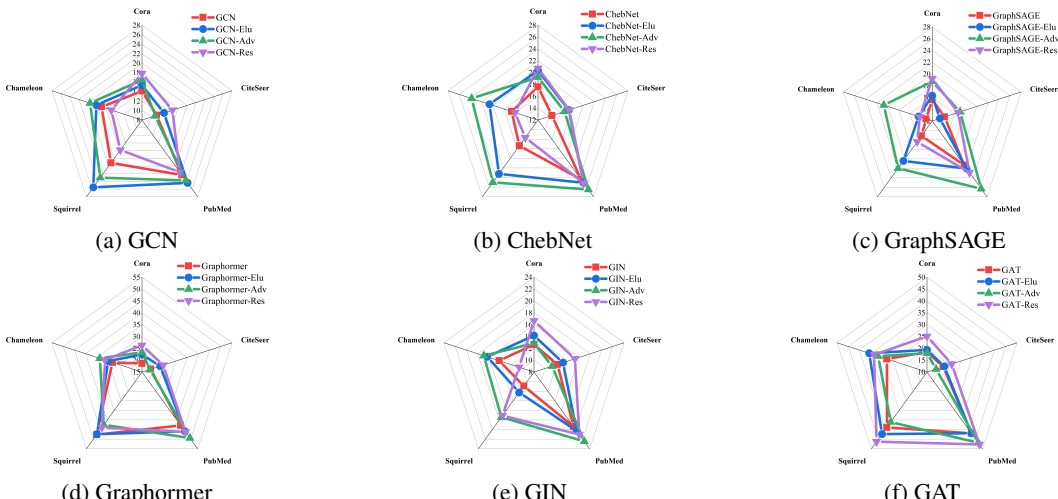

Figure 6: Comparison of three strategies for mitigating GNN over-invariance.

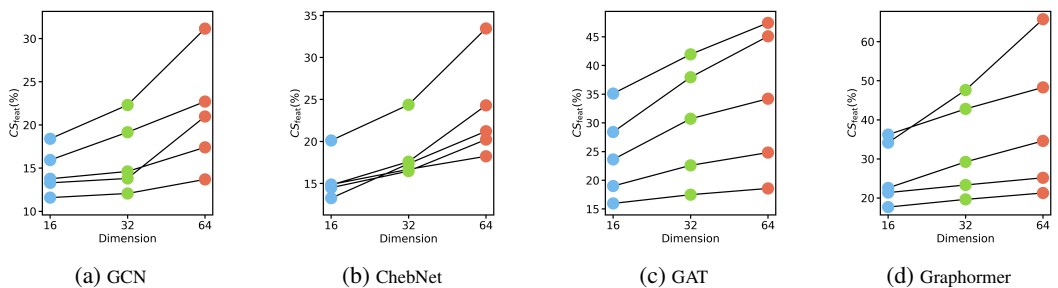

Figure 7: Comparison of hidden layer dimensions for mitigating GNN over-invariance.

### 4.5 LAYER-WISE FEATURE METAMER GENERATION

To investigate how metamer behavior changes across different layers of a model, we trained 4-layer GNNs and generated feature metamers by targeting activations at the 1st, 2nd, and 3rd layers, respectively. As shown in Figure 8, experiments across four models and five datasets reveal that the $CS_{\text{feat}}$ consistently decreases with increasing layer depth. Figure 9 visualizes the original PubMed features alongside feature metamers generated by targeting the 1st, 2nd, and 3rd layers of GAT. The similarity between the metamers and the original features drops noticeably as the targeted layer becomes deeper, but the activation similarity between the metamer and the reference graph in GAT remains as high as 98.71%.

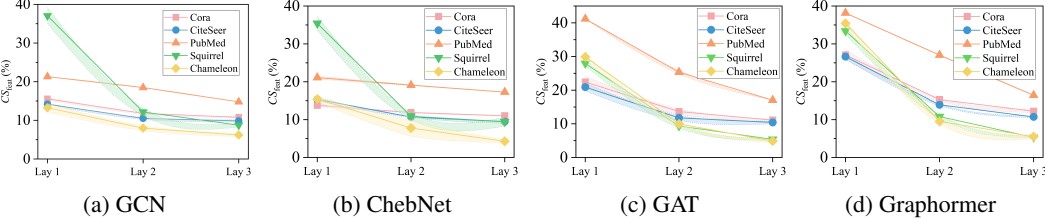

Figure 8: $CS_{\text{feat}}$ at different target layers.

Table 2: GNN classification accuracy (mean $\pm$ std) on cross-architecture feature metamers.

| Methods | Original | GCN | ChebNet | GraphSAGE | GIN | GAT | Graphormer |
|---|---|---|---|---|---|---|---|
| GCN | 78.35±0.24 | 78.40±0.30 | 77.43±0.12 | 76.32±0.28 | 76.38±0.49 | 78.32±0.31 | 76.08±0.35 |
| ChebNet | 75.79±0.96 | 77.05±0.51 | 77.31±0.82 | 76.82±0.41 | 75.60±0.26 | 77.03±0.44 | 76.24±0.74 |
| GraphSAGE | 76.71±0.27 | 76.13±0.40 | 76.17±0.23 | 75.38±0.70 | 76.21±0.29 | 74.73±0.37 | 77.35±0.41 |
| GIN | 76.22±1.01 | 77.06±0.37 | 77.12±0.71 | 77.02±0.42 | 75.46±0.37 | 76.64±0.42 | 76.38±0.61 |
| GAT | 75.75±0.96 | 73.54±1.86 | 76.39±0.48 | 71.14±2.32 | 76.30±0.12 | 73.37±1.32 | 73.17±1.95 |
| Graphormer | 76.43±0.31 | 75.78±0.58 | 75.65±0.46 | 74.16±0.68 | 76.28±0.46 | 74.57±0.33 | 76.51±0.59 |



a. $G$    b. $G'^{(1)}$    c. $G'^{(2)}$    d. $G'^{(3)}$

Figure 9: Feature metamers from different GAT layers (PubMed).

### 4.6 Evaluating Invariance Compatibility Between GNNs

In this experiment, we first trained seven representative GNNs on the PubMed dataset and recorded their baseline classification accuracy on the original graphs. To assess cross-model generalization, each model was then retrained using the feature metamers generated by every other model, and evaluated on the original test set. The resulting classification accuracies are reported in Table 2.

The results show that while each model can generally classify its own metamers reliably (diagonal entries), there are substantial differences in cross-model transferability. Notably, ChebNet exhibited consistently high robustness when classifying metamers from other models, whereas GraphSAGE showed a dramatic drop in accuracy across most metamers. Metamers generated by GAT proved to be the most disruptive for all models, completely impairing GraphSAGE's performance in particular. These findings suggest that the invariances learned by GNNs comprise both shared components—e.g., ChebNet tends to preserve signals crucial across models—and architecture-specific features—e.g., the invariances captured by GAT are largely uninterpretable to other models. Together, these elements shape the unique representational decision landscape of each model.

## 5 Conclusion

In this work, we present a principled framework for generating model metamers in GNNs and use it to uncover and quantify the models' representational invariance. Our analysis reveals that widely used GNN architectures often exhibit excessive invariance, mapping structurally or semantically different graphs to nearly identical internal activations. This over-invariance reflects a misalignment between model perception and human intuition, and can mask critical differences in the input. Through both theory and experiments, we characterize the metamer manifold, propose metrics to assess feature- and structure-level invariance, and explore architectural and training modifications—such as ELU activations, residual connections, and adversarial training—that help mitigate over-invariance. We also show how network depth and width affect invariance, and how cross-model comparisons reveal shared and divergent inductive biases. Altogether, our findings expose a core challenge in GNN design and provide tools for benchmarking and improving representational sensitivity. As an initial step toward analyzing GNN invariance via metamers, future work can extend this approach to more graph tasks, broader model families, and improved metamer generation techniques.

### Reproducibility Statement

We aim for full reproducibility. Formal assumptions and complete proofs are provided in the Appendix. Implementation details are documented in the main text, together with an anonymous downloadable code package.

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

# A  APPENDIX

## A.1  THEORETICAL STUDIES OF METAMER MANIFOLDS

In this section, we develop theoretical foundations for the existence of metamer manifolds and their volume in relation to model properties. These insights will inform and support our subsequent experimental analysis.

### A.1.1  LOCAL METAMER DIMENSION FOR SINGLE NODE

Let $f : \mathbb{R}^d \longrightarrow \mathbb{R}^m$ be a smooth mapping that takes as input the feature vector obtained by aggregating the features of a node and its neighbors,

$$\tilde{x} = \sum_{u \in \mathcal{N}(v)} \alpha_{uv}\, x_u \ \in\ \mathbb{R}^d \tag{18}$$

Denote its output by:

$$h = f(\tilde{x}) \in \mathbb{R}^m \tag{19}$$

**Jacobian and rank.**  The Jacobian of $f$ at $\tilde{x}$ is the $m \times d$ matrix:

$$Df(\tilde{x}) = \begin{bmatrix} \dfrac{\partial f_1}{\partial \tilde{x}_1}(\tilde{x}) & \cdots & \dfrac{\partial f_1}{\partial \tilde{x}_d}(\tilde{x}) \\ \vdots & \ddots & \vdots \\ \dfrac{\partial f_m}{\partial \tilde{x}_1}(\tilde{x}) & \cdots & \dfrac{\partial f_m}{\partial \tilde{x}_d}(\tilde{x}) \end{bmatrix} \tag{20}$$

and we write $r = \mathrm{rank}\big(Df(\tilde{x})\big)$. If $\alpha_{uv}$ is or learnable, the Jacobian includes extra terms from $\partial \alpha_{uv}/\partial x$, making $r$ sensitive to both features and learned edge weights.

**Local metamer dimension.**  At the specific input $\tilde{x}_v$, the local dimension of the Metamer set:

$$\mathcal{M} = \{\, \tilde{x}' \in \mathbb{R}^d \mid f(\tilde{x}') \approx f(\tilde{x}_v)\} \tag{21}$$

is given by:

$$\dim_{\tilde{x}_v} \mathcal{M} = d - r \tag{22}$$

This result follows from the rank–nullity theorem Behrmann et al. (2019) applied to $Df(\tilde{x}_v)$: since its kernel has dimension $d - r$, there are exactly $d - r$ independent directions along which infinitesimal changes leave $f(\tilde{x}_v)$ nearly unchanged. Equivalently, the Jacobian rank $r$ measures the number of feature-space directions affecting the output, while $d - r$ quantifies the local Metamer degrees of freedom. Designing GNNs to increase $r$ directly reduces $d - r$ and thus shrinks the metamer manifold.

### A.1.2  ACTIVATION INDUCED VOLUME CHANGE

Consider the coordinate-wise activation mapping $\sigma \colon \mathbb{R}^m \ \to \ \mathbb{R}^m$. Its Jacobian is the diagonal matrix:

$$D\sigma(z) = \mathrm{diag}\left( \frac{\mathrm{d}\,\sigma(z_1)}{\mathrm{d}z_1},\ \ldots,\ \frac{\mathrm{d}\,\sigma(z_m)}{\mathrm{d}z_m} \right) \tag{23}$$

where $z \in \mathbb{R}^m$ is the pre-activation vector and $\sigma$ acts independently on each coordinate. So the singular values of $D_\sigma$ are $|\mathrm{d}\,\sigma(z_v)/\mathrm{d}z_v|$, and the local volume scaling factor is:

$$\det D\sigma(z) = \prod_{v=1}^{m} \mathrm{d}\,\sigma(z_v)/\mathrm{d}z_v \tag{24}$$

For ReLU Nair & Hinton (2010), $\mathrm{d}\,\sigma(z_v)/\mathrm{d}z_v = 1$ if $z_v > 0$ and $0$ otherwise, hence $\det D\sigma \in \{0, 1\}$: zeros correspond to complete collapse along those coordinates and introduce extra Metamer directions. More generally, any $0 < \mathrm{d}\,\sigma(z_v)/\mathrm{d}z_v < 1$ contracts volume in direction $v$, increasing local invariance.

## A.2   THE USE OF LLM

We used a large language model to polish the writing, including wording, clarity, and grammar. The model did not generate data, code, or analyses, and it did not alter technical content, equations, results, or conclusions. All scientific ideas and validations were performed by the authors.

