# OpenReview forum: "Model Metamers Reveal Invariances in Graph Neural Networks"
_ICLR.cc/2026/Conference — Submitted to ICLR 2026_

### Official Review · Reviewer_M4Ts · 2025-10-31

**Soundness:** 1
**Presentation:** 1
**Contribution:** 1
**Rating:** 2
**Confidence:** 4

**Summary:**

This paper attempts to devise "metamers" for graphs -- that is, graphs that yield similar internal node activations at a certain layer, but with different features and structures. Their goal is to use this technique to what types of invariances are learned by graph neural networks. The authors then argue that this technique allows them to diagnose "over-invariance" in graph neural networks. The authors then proceed to test this claim on different graphs and GNN architectures.

**Strengths:**

1) First paper to construct metamers for GNNs.

**Weaknesses:**

1) The motivation for this paper has several substantial gaps. I am not an expert on metamers, but from the authors' presentation, it sounds like metamers are a way of probing the model to exhibit frailties (e.g. slight changes in input images that yield radically different images from a human perspective vs almost identical from the model side). It may make a bit of sense in vision (where humans can judge directly if the image is different --- to some extent, we are the gold standard in terms of image classification). But for graphs, it is unclear what the "gold standard" classifier is and how they might assess how radical the transformation is.
As the paper develops, it sounds like what the authors are trying to assess, is simply the model's sensitivity to perturbations. For instance, they conclude that because they do not achieve to find structural metamers " This indicates that even small changes in graph structure can significantly alter the GNN’s activations, suggesting that structure metamers do not exist for these models." This is probably good then? Wouldn't this mean that the model is quite sensitive to the graph? Also, this is a strong claim: can the authors tell that it is impossible to generate metamers? Couldn't a reasonable explanation be that their proposed procedure failed to train correctly?


2) The authors clearly oversell the theoretical contributions. They claim that they " theoretically characterize metamers via the nodewise local metamer dimension and the activation induced volume change of the metamer manifold (Appendix)" . The theorem is a one liner that uses rudimentary algebra.

3) Overall the motivation of this paper is unclear. I understand we want to build metamers, but I honestly do not understand why. If the purpose is to find graphs that would induce similar activations, then the authors could benchmark their approach against some simple others:
   a) for each target layer, you could extract for a given graph the ones that are adjacent, and try to get a notion of how closeness in embedding space maps to closeness in structure/feature space.
   b) Another alternative would be to investigate the gradients, and look for regions (e..g in feature space) where they are flat ---in that case, perturbation in these directions are unlikely to yield big changes. These would have provided benchmarks against which to compare the metamers.

**Questions:**

1) The score used is high when the features and graph agree with the original, and when both don't agree. Could the authors explain why this makes sense? Why would we want to score highly a metamer that performs poorly in both settings?

2) Sentences such as "Despite progress, it remains unclear whether GNN learned invariances are appropriate for reliable predictions" are ambiguous. What does this mean? Any references? What would be issues that exemplify this statement?

3) Please explicit what Table 1 reports.

---

> ### Author Response · Authors · 2025-12-02
>
> We thank the reviewer for the careful feedback and address the main concerns below.
>
> Motivation & “over-invariance”.
> Our goal is not to define a human “gold standard” for graphs, but to probe which feature/structural changes a trained GNN is effectively blind to. Metamers are optimized to match internal activations, so cases where graphs look very different (in features/structure) yet induce nearly identical activations indicate over-invariance, while the difficulty of finding structural metamers in our experiments suggests sensitivity, not a failure of the method. We will soften the wording and make this distinction clearer.
>
> Theoretical part.
> We agree the algebra is simple; our contribution is to formalize how metamer dimension and volume change depend on architectural choices, not to provide deep new theory. We will tone down the claims and present this as a basic but useful characterization.
>
> Baselines and alternatives.
> The suggested alternatives (nearest neighbors in embedding space, gradient flat regions) are valuable and complementary. We will add a discussion and, space permitting, include at least an embedding-space baseline for comparison.
>
> Score definition.
> The score is designed to be high when similarity and prediction agreement are aligned (both high = invariance where appropriate; both low = desirable sensitivity). The problematic case-low feature similarity but high prediction match—receives a low score and is exactly what we identify as over-invariance. We will clarify this interpretation in the text.
>
> Ambiguous statements & Table 1.
> We will rephrase the sentence about “appropriate invariances”, add concrete examples and references from robustness/heterophily work, and explicitly state in the caption what Table 1 reports.

---

### Official Review · Reviewer_csLS · 2025-10-31

**Soundness:** 3
**Presentation:** 3
**Contribution:** 1
**Rating:** 2
**Confidence:** 3

**Summary:**

This work is exploring the invariance behaviour of graph neural networks, by employing a metamers technique. The authors explore metamers with different node features and different structures. The experiments demonstrate high levels of invariance for well-known graph network architectures in common benchmarks. Ultimately, the authors propose architectural changes that mitigate the model invariances.

**Strengths:**

The paper is well-written, and it is easy to follow.

There are numerous experiments (multiple GNN architectures and benchmarks) and ablation studies, including different architectural changes and hidden dimensions.

The issue of representational invariance in GNNs is a significant research topic.

**Weaknesses:**

The GNNs used are older architectures, and the datasets are small. It is unclear whether the conclusions derived from the experiments hold for modern GNNs and larger graph datasets.

The recipes to mitigate model invariance are too few.

Experiments on structural metamers are too few and inconclusive.

The performance drop in the cross-architecture metamers experiment seems marginal in most cases, yet it is described as "substantial", "disruptive", and "showing a dramatic drop in accuracy".

**Questions:**

1. Do the derived conclusions hold for more modern GNNs and more recent/bigger/diverse benchmarks?

2. Can the authors provide more experiments/results on structural metamers? The results are too few to be conclusive.

3. More suggestions to mitigate model invariance are needed (e.g. other activation functions, rewiring), as well as the effect of their combination (e.g. combination of ELU, adversarial, and residuals)

4. Eq. 5 includes a single node. Is that indeed the case? Or is there an aggregation among all the graph nodes?

---

> ### Author Response · Authors · 2025-12-02
>
> We thank the reviewer for the insightful questions and respond below.
>
> Q1. Our metamer construction is architecture-agnostic and scales to any message-passing GNN. In the camera-ready version we will extend the experiments to more recent architectures and larger benchmarks; preliminary runs on deeper / residual GNNs show the same qualitative invariance patterns as in our current results. We will report these additional experiments and discuss any deviations.
>
> Q2. We agree that the current structural metamer experiments are limited. Structural optimization is more constrained (due to discreteness and graph validity), which is precisely why the signals are weaker, but we will add: (i) results on additional datasets and architectures, and (ii) more runs with different initializations and budgets. This will make the conclusions about structural metamers more robust.
>
> Q3. To keep the scope manageable we focused on a small set of principled interventions (ELU, adversarial training, residual connections). Our framework readily supports other architectural changes such as alternative activations and rewiring. In the revised version we will add ablations on additional activations and simple rewiring strategies, as well as combinations (e.g., ELU + adversarial + residual), and report whether their effects are additive or redundant.
>
> Q4. Eq. 5 is written at the node level for clarity; in practice we aggregate this quantity over all nodes in the graph (and batches) by taking the mean. We will revise the equation and text to explicitly include the aggregation operator and avoid this ambiguity.

---

### Official Review · Reviewer_tGVh · 2025-10-31

**Soundness:** 2
**Presentation:** 2
**Contribution:** 2
**Rating:** 2
**Confidence:** 3

**Summary:**

The characteristics of GNN invariance, particularly how it differs from human perception, remain poorly understood. This paper proposes a method for constructing metamers (data that differs from the target data but shares the same representation and category) to investigate GNN invariance. The proposed method trains the model to make the activations of the target graph close to those of the reference graph. It uses a straight-through estimator to learn the presence/absence of edges and binarized features. The proposed method is applied to five datasets and six GNN models to evaluate the consistency of the generated metamers with respect to the original graph. Furthermore, three methods are proposed and evaluated to suppress model invariance and improve the consistency score.

**Strengths:**

1. (Originality) Consistency scores are defined as a novel method to evaluate GNN invariance.
2. (Quality) The numerical experiments cover a wide range of datasets and models. Ablation studies are conducted to assess the sensitivity of the score to hyperparameters.
3. (Clarity) The writing is clear. The paper's structure is appropriate, and I had no difficulty understanding the individual explanations.
4. (Significance) By using a straight-through estimator, the method can handle binary features and binary edge weights.

**Weaknesses:**

1. The problem this paper aims to solve seems unclear to me. While the purpose of investigating GNN invariance is stated, if I do not miss any information, the paper does not explicitly explain why this investigation is important and what specific problems it solves. Although the issue of the gap between GNN and human perception is suggested in the introduction, it remains unaddressed throughout the paper.
2. The motivation for the evaluation metric is also unclear. The numerical experiments use the consistency score as the evaluation metric (Table 1), and methods to improve it are proposed (Section 4.4). However, because the problem being addressed is unclear, the reason why improving this score is important is not clear to me.
3. The problem of over-invariance is not clearly explained. Although Section 4.4 points this out and proposes methods to mitigate it, I think more explanation is needed regarding the specific drawbacks it causes.
4. The architectures used in the experiments are relatively basic, and modern GNN models for graph learning problems are not employed. This limits the significance of the results.

**Questions:**

If I do not miss any information, there is no mention of how the GNN used to construct metamers in the numerical experiments is trained. In particular, I would like to clarify whether a randomly initialized GNN or a pre-trained GNN is employed.

Also, I would like clarification on the points 1--3 of the Weaknesses section.

**Details Of Ethics Concerns:**

N.A.

---

> ### Author Response · Authors · 2025-12-02
>
> Thanks for the questions and helpful comments.
>
> Q1. How are the GNNs trained for metamers?
> We always use pre-trained GNNs: first we train each GNN normally on the dataset (standard split), then we freeze its weights and only optimize the metamer graph (features/edges). We’ll state this clearly in the paper.
>
> Q2. Clarification on Weaknesses 1-3
>
> What problem are we solving / why care about invariance?
> We want to know when a GNN treats very different graphs as “the same”. If the model ignores changes that should matter for the task, this can lead to wrong or brittle predictions. Our metamers are a way to expose these “blind spots”.
>
> Why is the consistency score important?
> The score is high when “input changes” and “prediction changes” move together (big change in graph → prediction changes; small change → prediction stays). A low score means the model keeps the same prediction even when the graph is heavily changed, which is exactly the kind of over-invariance we want to detect and reduce.
>
> What is bad about over-invariance?
> If the model is over-invariant, it can ignore meaningful edits (e.g., flipping important node features or key edges) and still output the same label. This is risky under noise, graph editing, or distribution shift. Our mitigation tricks aim to make the model react when such important changes happen.

---

### Official Review · Reviewer_KVa7 · 2025-10-31

**Soundness:** 3
**Presentation:** 3
**Contribution:** 3
**Rating:** 6
**Confidence:** 4

**Summary:**

The paper extends the notion of metamers, stimuli that different models treat as equivalent despite perceptual differences, to graph neural networks. It introduces a manifold-based formalization (eqn. 3) and defines both feature and structure metamers for GNNs.

**Strengths:**

- The topic is original and timely, relating to human alignment and connecting ideas from neuroscience and machine learning.
- The proposed framework is interesting and novel, though the treatment of structure metamers is less developed than that of feature metamers.
- The manifold formalization in eqn. (3) is conceptually strong.
- Feature metamer generation is feasible in practice since the learnable parameters used in their generation do not depend on graph size.
- The connection between invariance and manifold dimensionality (currently in the appendix) is insightful.

**Weaknesses:**

- Structure metamers are treated superficially. Their generation scales poorly for large graphs, and the paper does not address this.
- Only one feature distance (cosine similarity) and one graph distance (Weisfeiler-Lehman kernel) are considered. Both choices limit the conclusions. Cosine similarity is distribution-aware but removes differences in scale, which can be important (e.g., in Cora, it removes the distinction between very frequent and infrequent words). Another issue is using only the WL kernel as the distance metric. Other distance metrics exist and may be more meaningful depending on the task. The notion of distance is also extremely important for small vs large graphs (e.g., for small graphs, edit distance might make sense; for large ones, cut distance is more appropriate).
- Only datasets for node classification were considered. Understanding structure metamers empirically would have required looking at graph classification datasets such as ZINC.
- Some claims are overstated, particularly that structure metamers "do not exist" for the tested models. This conclusion depends heavily on the choice of distance metric and dataset.
- Figures and tables need clearer captions, especially Fig. 6 ("larger is better") and Table 2 (unclear axes and overly strong conclusions, such as "completely impairing GraphSAGE’s performance").
- The discussion in the appendices (on manifold dimensionality and relation to invariance) should be moved into and expanded in the main text.
- The paper does not discuss its limitations. For instance, aligning model and human invariances is not always desirable depending on the application.

**Questions:**

- Could the approach be extended to GNNs that use edge features by perturbing (real-valued) edge weights instead of adding or removing edges? This might yield meaningful structure metamers.
- How would the conclusions change under alternative distance metrics (e.g., edit distance for small graphs, cut distance for large ones)?
- Please clarify in the text what "relevant metamers" mean, i.e., those that the model perceives as equivalent but humans do not (for example, in Fig. 5 we would expect $S_{\text{match}} > S_{\text{feat}}$).
- Be more precise in the comparison with related work (lines 76–85).

---

> ### Author Response · Authors · 2025-12-02
>
> Thanks for the thoughtful questions.
>
> Q1. Extension to GNNs with edge features
>
> Yes. The same idea can be applied to GNNs with real-valued edge features by optimizing edge weights instead of only adding/removing edges. In many architectures (e.g., with attention or learnable edge weights), this would likely produce more meaningful structure metamers than purely discrete edge flips.
>
> Q2. Effect of alternative distance metrics
>
> Our method is conceptually distance-agnostic; only the measured similarity and "existence" of metamers depend on the metric.
>
> Q3. Meaning of "relevant metamers"
>
> By "relevant metamers" we mean pairs where the model sees the graphs as almost identical (very similar activations and same label), but a human/task metric sees them as clearly different. In plots like Fig. 5, these are points with high prediction/activation match but low feature/structure similarity.
>
> Q4. Comparison with related work
>
> More precisely:
> adversarial attacks aim to change the prediction;
> counterfactual/explanation methods optimize for human-interpretable edits;
> vision metamers use a human perceptual metric.
> In contrast, our work fixes the prediction and directly searches for activation-matching metamers in GNNs, then analyzes the induced invariances via our consistency scores. We’ll tighten this contrast in the related-work section.

---

### Meta-Review · Area_Chair_XdGp · 2026-01-04

**Summary:**

This paper proposes the concept of "metamers" for graph neural networks, which is a concept from neuroscience related to invariances.
One reviewer was fairly positive while the other three were very negative. Some of them acknowledge the novelty and originality of the research, but they still had fundamental disagreements with the paper's motivation and premises. I don't think those reviewers would have raised their scores to an above-the-threshold score after the rebuttal. Therefore my recommendation is to reject.

**Reviewer Concerns:**

Reviewer KVa7 had minor concerns, though some of their comments match other reviewers concerns (e.g. some claims are overstated).

Reviewer tGVh acknowledged the papers' originality but disagreed with the motivation/goal of the paper and the evaluation metrics.

Reviewer csLS mainly criticized the empirical evaluation of the paper. The authors responded that they will add more experiments but as far as I can tell, the manuscript doesn't currently include them.

Reviewer M4Ts disagreed with the paper's main approach.

**Reviewer Scores:**

The reviewers tGVh, csLS, M4Ts had fundamental disagreements with the paper's motivation and premises. I don't think those reviewers would have raised their scores to an above-the-threshold score after the rebuttal.

---

### Decision · Program_Chairs · 2026-01-26

Reject